# Inclined Obstacle Recognition and Ranging Method in Farmland Based on Improved YOLOv8

Xianghai Yan [1,2], Bingxin Chen [1], Mengnan Liu [2], Yifan Zhao [1] and Liyou Xu [1,2,*]

1 College of Vehicle and Traffic Engineering, Henan University of Science and Technology, Luoyang 471003, China; 9905167@haust.edu.cn (X.Y.); 210321030321@stu.haust.edu.cn (B.C.); 220320030288@stu.haust.edu.cn (Y.Z.)

2 State Key Laboratory of Intelligent Agricultural Power Equipment, Luoyang 471000, China; liumengnan27@163.com

* Correspondence: xlyou@haust.edu.cn

**Abstract:** Unmanned tractors under ploughing conditions suffer from body tilting, violent shaking and limited hardware resources, which can reduce the detection accuracy of unmanned tractors for field obstacles. We optimize the YOLOv8 model in three aspects: improving the accuracy of detecting tilted obstacles, computational reduction, and adding a visual ranging mechanism. By introducing Funnel ReLU, a self-constructed inclined obstacle dataset, and embedding an SE attention mechanism, these three methods improve detection accuracy. By using MobileNetv2 and Bi FPN, computational reduction, and adding camera ranging instead of LIDAR ranging, the hardware cost is reduced. After completing the model improvement, comparative tests and real-vehicle validation are carried out, and the validation results show that the average detection accuracy of the improved model reaches 98.84% of the mAP value, which is 2.34% higher than that of the original model. The computation amount of the same image is reduced from 2.35 billion floating-point computations to 1.28 billion, which is 45.53% less than the model computation amount. The monitoring frame rate during the movement of the test vehicle reaches 67 FPS, and the model meets the performance requirements of unmanned tractors under normal operating conditions.

**Keywords:** unmanned tractor; inclined working condition; dedicated dataset; algorithm improvement; camera ranging

## 1. Introduction

Obstacle recognition is a core aspect of vehicle assisted driving control systems [1–3]. As assisted driving systems for passenger cars are becoming more and more mature, they are gradually being applied to intelligent agricultural machines. Compared with passenger cars, tractors always have one side of their wheels stuck in the soft soil after ploughing, so that the vision sensor always has an inclination of about 10° [4], as shown in Figure 1. This affects the detection of obstacles. In addition, violent shaking of the tractor can reduce the accuracy of the camera in recognizing obstacles in the field [5]. The hardware cost of the tractor's assisted driving system and the lack of controller arithmetic are also key technologies that need to be solved urgently.

Current visual perception is mainly based on deep learning methods for detection [6], which can be divided into One-stage and Two-stage according to the detection stage [7]. R-CNN series [8–10] is a very classical Two-stage algorithm, which firstly extracts the candidate frames, then filters them using a classifier, and finally removes the duplicate frames and fine-tunes the predicted frames using non-maximal suppression [11]. Two-stage has some advantages in terms of detection accuracy, but suffers from the disadvantages of difficult training, slow detection, and difficult optimization. Single-stage detectors, including methods such as YOLO [12–17] and SSD [18], benefit from the excellent performance of the Transformers [19] detection model in the field of natural language processing, and

have a faster detection speed. Debasis Kumar et al. [20] demonstrated that training with a specially customized dataset can significantly improve the detection performance of YOLOv8; Wang Zhibin et al. [21] demonstrated that the larger the self-constructed dataset, the better the training and the vehicle detection accuracy of the YOLO algorithm. Sun Zhongzhen et al. [22] expanded the tilted dataset to improve the model's recognition accuracy of tilted obstacles. All of the above studies proved that training vision algorithms with self-constructed field obstacle-specific large datasets can improve the detection of field obstacles by tractor assisted driving systems.

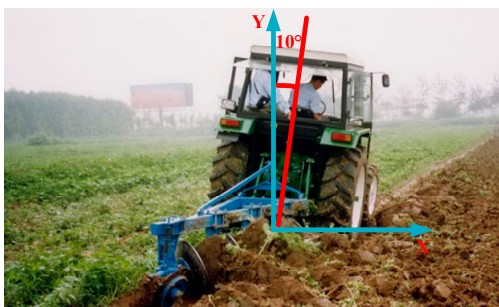

**Figure 1.** Tractor Inclination Diagram in Plowing Condition.

In 2020, the team of Ningning Ma [23] proposed the activation of features using the FReLU function, which achieves the pixel-level modelling capability in a simple way; scholars such as Ying Yao [24] proved the pixel-level modelling capability of the FReLU function. In 2017, the MobileNet [25] model was proposed, and this network can significantly reduce the amount of model computation while maintaining the performance of the model. Numerous scholars applied the MobileNet to the YOLO algorithm and verified the contribution of MobileNet to model lightweighting [26–28]. Mingxing Tan and others [29] proposed a bidirectional weighted feature network BiFPN, which can scale the depth and width of the backbone network and feature network. Yunfei Zhang et al. [30] proposed a multi-scale image feature fusion compression algorithm using arithmetic coding to eliminate the statistical redundancy of compact features, which solves the problem of the large amount of feature data, which makes it difficult to transmit efficiently. Guo Yue et al. [31] used BiFPN for a YOLO model to significantly reduce the number of floating points. Fan Zhang [32] et al. applied the improved attention module to the deep learning model to significantly reduce the number of parameters of channel attention and improve the detection speed. Bisong et al. [33] used a monocular camera to complete obstacle pose estimation and localization, saving costs but with lower accuracy. Jingliang et al. [34] used a binocular camera to localize pedestrians in orchards to complete high accuracy pedestrian monitoring. Although these studies have improved the detection accuracy of vision algorithms and reduced the cost, they are only applicable to obstacle detection under a normal viewing angle. In this paper, the research focuses on solving the problem of accuracy reduction caused by the tilted perspective and violent shaking of the tractor under ploughing conditions.

The purpose of the improved model proposed in this paper is to reduce the effect of tractor body tilt on the obstacle recognition effect, and at the same time save the hardware cost and controller computational resources of the tractor's unmanned or assisted driving control system. In this paper, the YOLOv8 algorithm is improved in the following aspects: firstly, the binocular ranging module is added to the algorithm, so that the ranging task can be accomplished without the use of radar; then, the backbone network is improved by using MobileNetV2 to lighten the backbone network; then, the neck network is improved by using BiFPN to lighten the neck network; then, by improving the activation function and embedding the SE noticing mechanism, the model accuracy can be improved; and finally, a specialized dataset is established to improve the accuracy of obstacle detection

in tilted view. The task of monitoring obstacles in the field with high accuracy is realized while the model is lightweighted.

### 1.1. Activation Function Improvement

The YOLOv8 algorithm classifies and identifies field obstacles through neural networks, and the binocular ranging algorithm outputs the distance of field obstacles using the binocular parallax principle, and the two combine to complete the monitoring task. The principle of the whole monitoring model is shown in Figure 2. The backbone network of the YOLOv8 model is the CSPDarkNet [35] model, which has a large number of convolutional modules and needs to occupy a large amount of controller arithmetic. In this paper, the convolution module is optimized with MobileNetV2, which can reduce nearly half of the computation without changing the number of feature maps and save the computational resources of the unmanned tractor. At the same time, the SE attention mechanism [36] is embedded in the backbone network, which improves the field obstacle detection accuracy while reducing the computational volume. The activation function of YOLOv8 is the SiLU activation function [37], which is less robust and the recognition confidence polarization is serious in the process of tractor field operation. In this paper, we adopt the FReLU activation function with stronger robustness, which is more suitable for the actual operating conditions of the tractor. The neck network of YOLOv8 model has two fewer convolutional connection layers than the previous feature pyramid network FPN [38], and it is not easy for the large feature maps to be fused with the small feature maps, so this paper uses the BiFPN path aggregation network, which fuses feature maps at different scales. The detection accuracy is further improved while saving the controller computational resources.

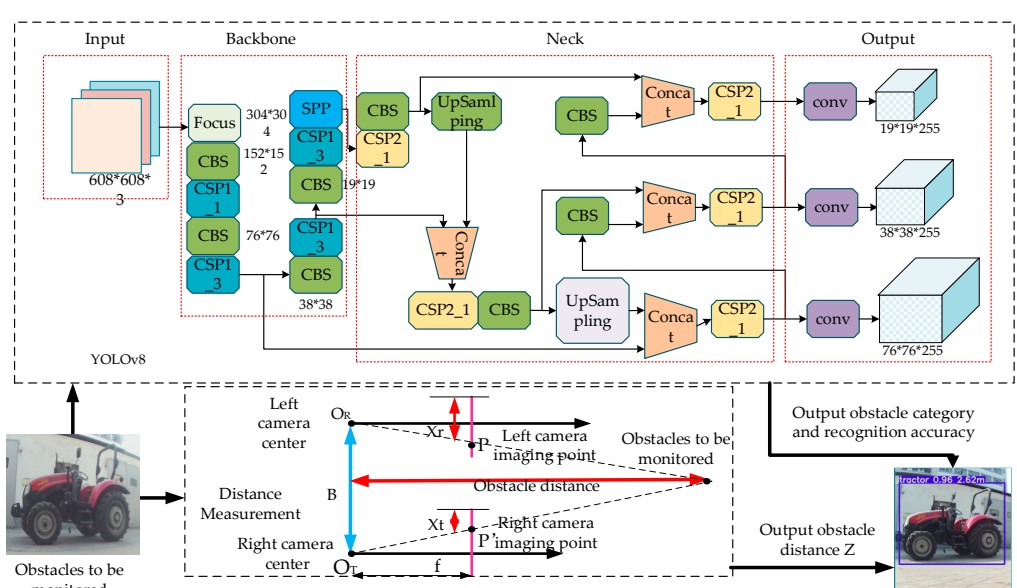

**Figure 2.** Monitoring model.

In the case of tractor ploughing, the tractor body will shake violently and the body will be tilted to the right by about 10°, the SiLU activation function space used in the original YOLOv8 is not sensitive, and the recognition error of tilted obstacles is more than 10%. As shown in Figure 3a, the confidence level of the original model is 96% at the highest and 46% at the lowest, with an error as high as 50%; as shown in Figure 3b, the confidence level of this model is 98% at the highest and 89% at the lowest, with an error of 9%. This is due to the fact that the stability of the SiLU activation function is not good enough, and in the picture appears slightly tilted. The recognition effect will be unstable, in contrast, because this model uses the FunnelReLU activation function, which makes the model more

resistant to interference. Even when the tractor is ploughing, the camera is often jittery, and the recognition effect of the obstacle will not see great change.

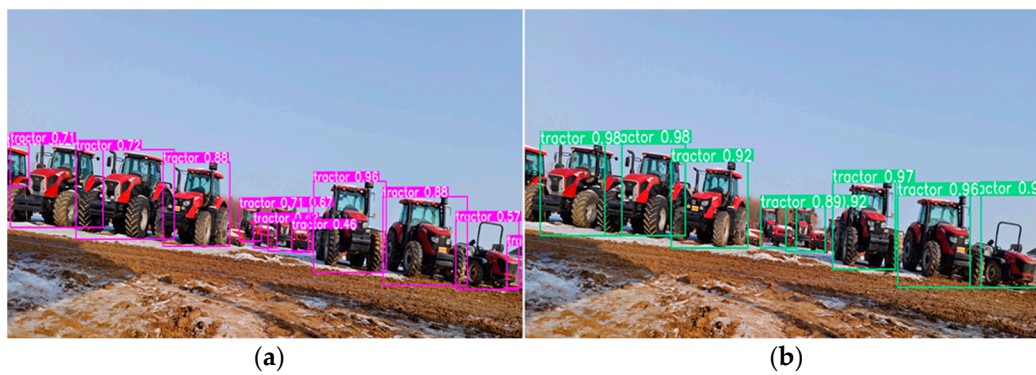

**Figure 3.** (**a**) SiLU Detection Performance; (**b**) FunnelReLU Detection Performance.

FunnelReLU has two core components. The first one treats the non-positive part of the original activation function with zeros by adding a funnel condition as in Equation (1). FunnelReLU extends it to a 2D condition that depends on the spatial context of each pixel. The spatial pixel calculation is shown in Equations (2) and (3).

$$y = \max(x, 0) \tag{1}$$

$$f(x_{c,i,j}) = x_{c,i,j}^{\omega} \cdot p_c^{\omega} \tag{2}$$

$$T(x_{c,i,j}) = x_{c,i,j}^{\omega} \cdot p_c^{\omega} \tag{3}$$

$T(x_{c,i,j})$—the funnel condition, $x_{c,i,j}^{\omega}$—the window on c channel centered at 2D position $(i, j)$, i and j denote the co-ordinate positions of the center window, $p_c^{\omega}$—the parameters shared by this window in the same channel.

The second place extends the original activation function to a sizable activation function with pixel-level modelling capabilities by using spatial conditions in the activation function through a convolution operation. This allows the network to generate spatial conditions for each pixel in the nonlinear activation, solving the spatial insensitivity problem. The activation region of FunnelReLU can be not only rectangular, but also rectangular or curved, which makes a better match with the shape of the object itself. A comparison of activation regions of FunnelReLU, ReLU and PReLU [39] is shown in Figure 4. On the rightmost side of the figure is the original ReLU function without any modifications and additions, in the middle is the FReLU activation function with common parameter conditions added on top of the ReLU activation function, and on the rightmost side is the FunnelReLU activation function with common parameter conditions added on top of the ReLU activation function.

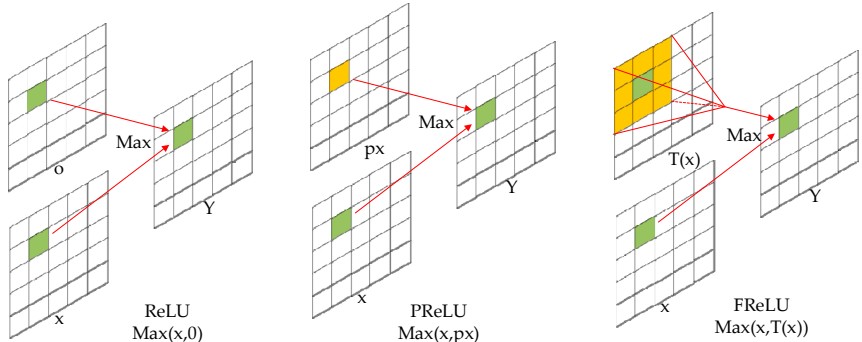

**Figure 4.** Activate function comparison.

*1.2. Feature Map Output Method Improvement*

The backbone network of YOLOv8 consists of convolutional layers, and the convolutional module consumes a lot of controller computational resources. MobileNetV2 [40] first enlarges the dimension, which does not increase a lot of computation, but helps the network to better learn and express the image features, then extracts the features with depth-separated convolution, and finally compresses the data with a projection layer, which effectively solves the problem of the model dimension and latitude of the model. This structure also saves controller computational resources without changing the number of model dimensions. After replacing the convolution module with MobileNetV2 in the paper, the floating point computation of the same image is reduced from 2.35 billion times to 1.28 billion times, which reduces the floating point computation by 45.53% without changing the number of feature maps and saves the computational resources of the controller of the tractor. The number of layers of the MobileNetV2 network is shown in Table 1.

**Table 1.** MobileNetV2 network layer.

| Input | Operator | t (Expansion Factor) | c (Output Channels) | n (Repeats) |
|---|---|---|---|---|
| $224^2 \times 3$ | conv2d | - | 32 | 1 |
| $112^2 \times 32$ | bottleneck | 1 | 16 | 1 |
| $112^2 \times 16$ | bottleneck | 6 | 24 | 2 |
| $56^2 \times 24$ | bottleneck | 6 | 32 | 3 |
| $28^2 \times 32$ | bottleneck | 6 | 64 | 4 |
| $14^2 \times 64$ | bottleneck | 6 | 96 | 3 |
| $14^2 \times 96$ | bottleneck | 6 | 160 | 3 |
| $7^2 \times 160$ | bottleneck | 6 | 320 | 1 |
| $7^2 \times 320$ | conv2d $1 \times 1$ | - | 1280 | 1 |
| $7^2 \times 1280$ | avgpool7 $\times$ 7 | - | - | 1 |
| $1 \times 1 \times 1280$ | conv2d $1 \times 1$ | - | k | - |

*1.3. Embedded Attention Mechanism*

The YOLOv8 model has different utilization rates for different channels but the same weights in the pooling process, resulting in computational loss and accuracy degradation. To further improve the detection accuracy of unmanned tractor, the SE attention mechanism is embedded in the YOLOv8 model. The SE attention mechanism consists of three parts, namely, squeezing, excitation and combination, as shown in Figure 5. The compression operation is a global average pooling [41] operation, which compressed the feature map from W × H × C to a 1 × 1 × C vector map, and the excitation part consists of two fully connected layers [42] and an activation function The two fully connected layers are composed of two fully connected layers for the number of neurons to reduce the latitude and then increase the dimension, and the activation function multiplies the channel weights to complete the model excitation. The excitation part is formulated as follows.

$$Z_c = F_{sq}(u_c) = \frac{1}{HW} \sum_{i=1}^{H} \sum_{j=1}^{W} u_c(i,j) \tag{4}$$

$$s = F_{ex}(z, W) = \delta(g(z, W)) \tag{5}$$

The meanings of the letters in the formula are as follows: δ—ReLU function, $W_1W_2$—two fully connected layers, z—combination operation.

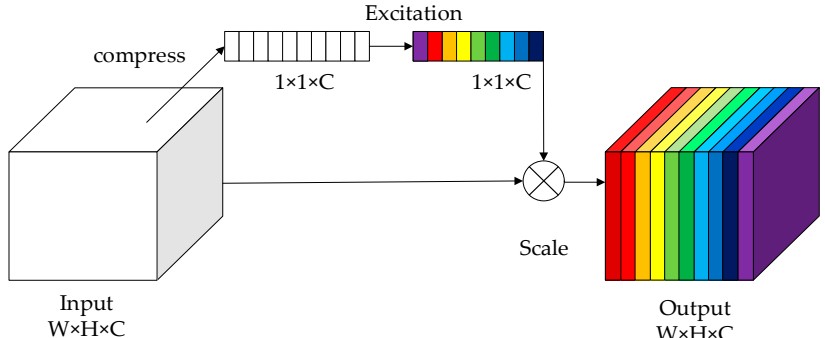

**Figure 5.** SE Structure diagram.

*1.4. FPN Optimization*

The PAnet [43] used in the neck network in YOLOv8 is the simplest bidirectional feature fusion, and its edge-most output routes contribute less to the feature network, which wastes the controller computational resources. In the paper, BiFPN is used to remove the edge-most two fusion routes and connect them to other routes in order to realize a higher-level feature fusion, and the routes are shown in Figure 6.

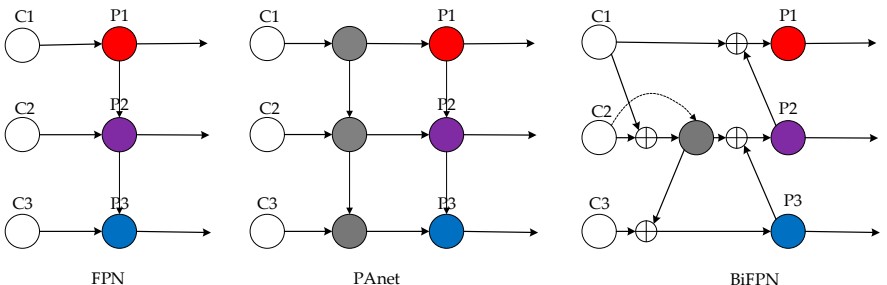

**Figure 6.** Comparison of feature fusion models.

The original feature fusion in YOLOv8 only superimposes feature maps, but different feature maps have different resolutions, and their contributions to the output features are different. The BiFPN feature fusion mechanism introduces different weighting coefficients to the fusion routes of the feature maps according to the different resolutions, which is able to better balance the feature information of different scales. The comparison of the three weighting methods is as follows.

$$O = \sum_i \omega_i \cdot I_i \tag{6}$$

$$O = \sum_i \frac{e^{\omega_i}}{\sum_j e^{\omega_j}} \cdot I_i \tag{7}$$

$$O = \sum_i \frac{\omega_i}{\epsilon + \sum_i e^{\omega_j}} \cdot I_i \tag{8}$$

The weighting method of Equation (6) is to directly add a learnable weight, but the weight is not limited, which may cause instability in training; the weighting method of Equation (7) can scale the weight range to between 0 and 1, which is stable, but the training is very slow; the weighting method of Equation (8) is similar to Softmax [44] that can scale the range to between 0 and 1, and the training efficiency is high. In this paper, the weighting method of Equation (8) is used to extract the layers 5, 7, and 11 of the backbone network to the neck network, and compress and merge the number of its channels into the BiFPN network for multi-scale feature fusion, and finally output to the detection end.

*1.5. Add Binocular Ranging Mechanism*

Binocular ranging uses the parallax principle to calculate the coordinates of the target point in three-dimensional space and consequently the target distance. The principle is

shown in Figure 7, and the obstacle distance Z can be obtained by only obtaining the parallax information D (Xr − Xt).

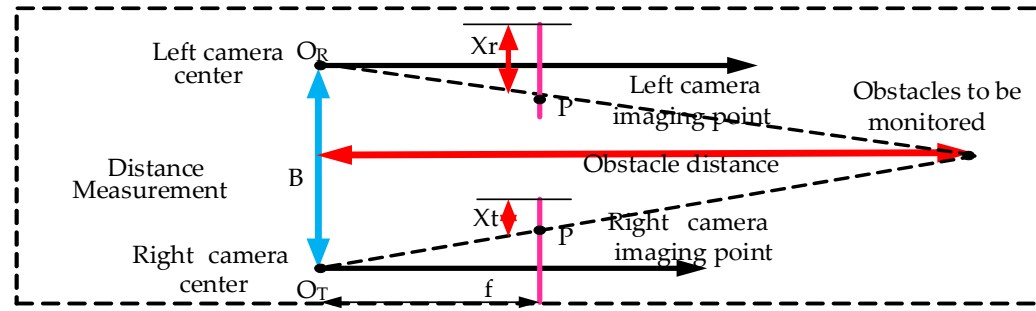

**Figure 7.** Binocular ranging principle.

The steps of binocular ranging are as follows: firstly, the focal length, distortion coefficient, rotation matrix and translation vector and other parameters of the camera are obtained through calibration; then, the same field obstacle is photographed with the left and right cameras, and thereafter the feature points are extracted from the left and right images and matched to obtain the distance of the field obstacle relative to the tractor. The binocular ranging effect of this paper is shown in Figure 8. The left side of the figure is the main camera view, which shows the confidence level and obstacle distance; the right side is the auxiliary camera view, which only shows the confidence level. All the following sections use only the main camera viewpoint information to show the model effect.

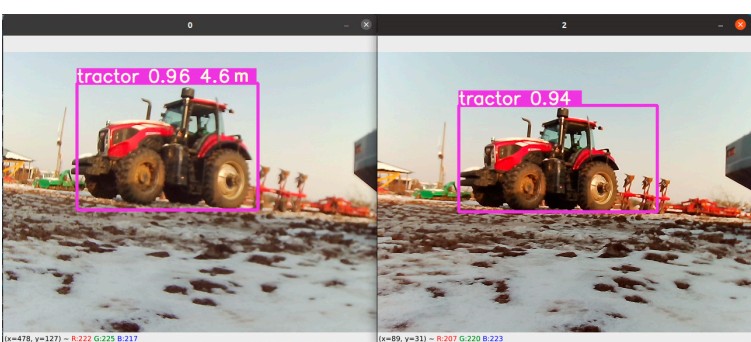

**Figure 8.** Stereo ranging effect.

## 2. Model Training

### 2.1. Experimental Platform

The code optimization of the experiment is based on Python3.8 and the CUDA10.2 framework, and the model training is based on the Colab Pro+ platform of Google cloud hard drive. The model training is completed by applying Google cloud GPU. The experimental training batch size is Batch = 64, Momentum = 0.9, Learning Rate Initial lr = 0.001, and the number of training iterations Epoch = 500.

### 2.2. Experimental Data Set

In order to simulate the obstacle images in the inclined view of the tractor, this paper uses a homemade dataset. Baidu unmanned vehicles were used to collect obstacle images from farmland in Xin'an County, Luoyang City and Cuigou test field of China YTO Group, and a total of 891 field obstacle images were collected to make the original dataset. Then, add the image rotation function to the original dataset to obtain obstacle pictures with different tilt angles for training, in order to simulate the obstacle perspective after the tractor's unilateral wheel is trapped under the real ploughing conditions. Afterwards, add mirror reversal function to simulate taking pictures from different sides of an obstacle;

add lighting to simulate the obstacle style under different sunlight; and build a dataset of 3500 obstacle pictures. The dataset is randomly divided into training set, test set and validation set according to the ratio of 7:2:1. Finally, Make sense was used to manually annotate the dataset, and three categories of trees, pedestrians and tractors were set.

### 2.3. Experimental Results and Analysis

In the paper, four performance metrics—mAP (mean average precision), precision, recall, and FPS (frames per second)—were used to evaluate the performance of the improved algorithm.

$$AP = \sum_{i=1}^{n-1} (r_{i+1} - r_i) P_{inter}(r_i + 1) \tag{9}$$

$$map = \frac{\sum_{i=1}^{k} AP_i}{k} \tag{10}$$

$$Precision = \frac{TP}{TP + FP} \tag{11}$$

$$Recall = \frac{TP}{TP + FN} \tag{12}$$

The meanings of the letters in the formulae are shown below: TP denotes the number of positive categories judged correctly, FP denotes the number of positive categories judged incorrectly, FN denotes a positive category judged as a negative category, and $r_i$ are the recall values corresponding to the first interpolated value at the first interpolated value in the precision interpolation segments arranged in ascending order. The overall accuracy of all categories of data is synthesized into mAP values. The parameters of the trained model are shown in Figure 9. The experimental results show that the improved YOLOv8 model has a mAP value of 98.84%, a recall value of 95.81%, a precision value of 97.90%, and a reduction in computation for the same image from 2.35 billion floating-point computations to 1.28 billion computations, which improves the average detection precision by 2.34% and reduces the amount of computation by 45.53%.

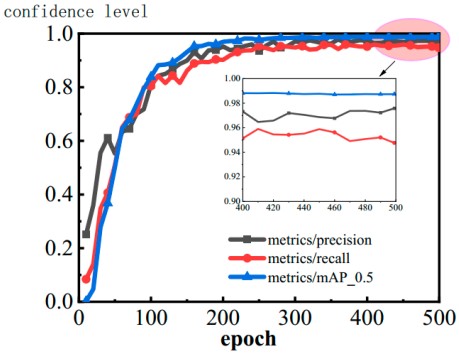 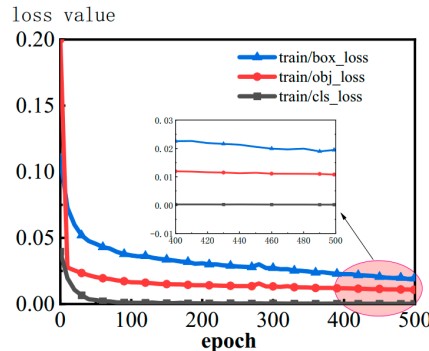

**Figure 9.** Training result.

### 2.4. Comparison Test

In order to verify the performance of the present model, a real-view comparison test is carried out at the Cuigou test field of China YTO Group under the right-tilt 10° view angle of an unmanned tractor. The present model is compared with the mainstream model, and different algorithms are used to train the same field obstacle dataset. The detection accuracy of the same live view obstacle is compared after the training is completed, and the results are shown in Figure 10. The test groups (a), (b), (c) and (d) in the figure are the present model, YOLOv8s, YOLOv7s and YOLOv5s models, respectively, and the results of the comparison test can be obtained as follows: compared with the other models, the present model has a low false-negative rate, high average confidence, and the confidence error of the identification of multi-obstacle is within 10%. Compared with the original model, the detection accuracy and robustness of the present model have been substantially improved.

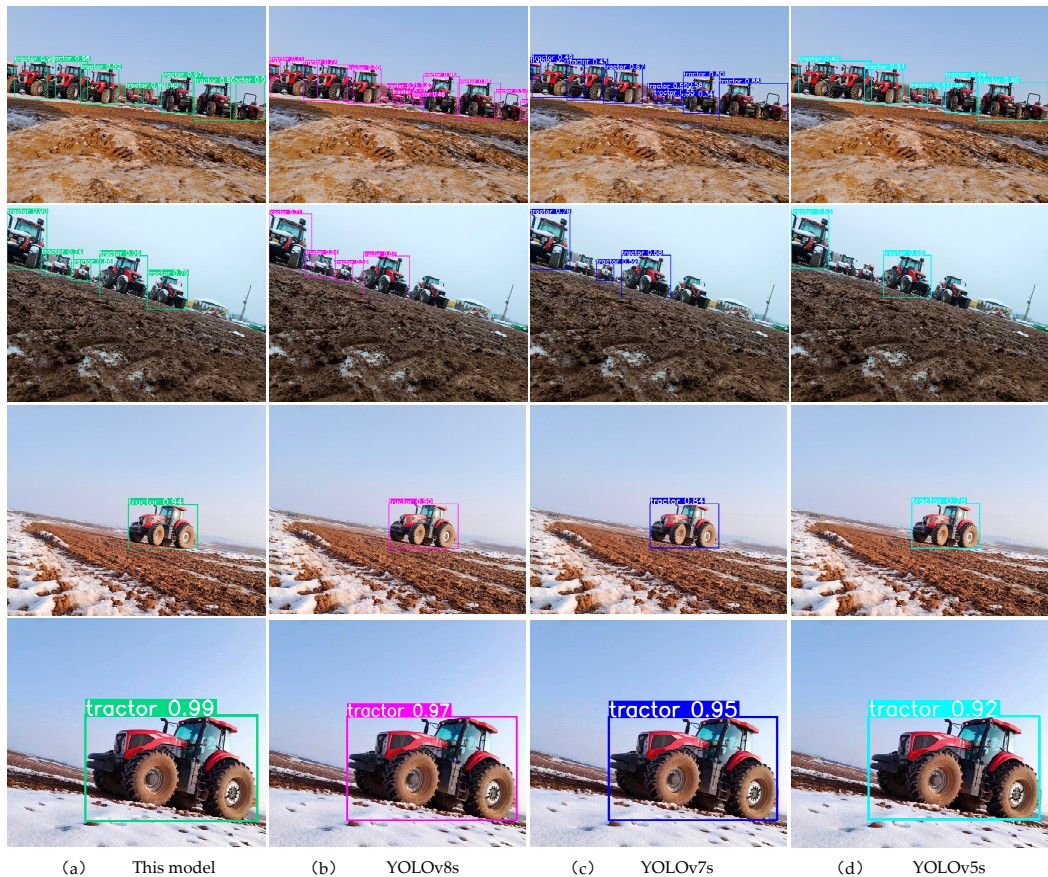

| (a) | This model | (b) | YOLOv8s | (c) | YOLOv7s | (d) | YOLOv5s |

**Figure 10.** Comparative test.

## 3. Real Vehicle Verification

In this paper, a robotic vehicle equipped with a mega2560 embedded microcontroller and two 12 mm focal length cameras is used for the tests. The specific parameters of the test equipment are shown in Table 2.

**Table 2.** Test platform parameters.

| Device Name | Deep Learning Model Robot (RobotCAR) |
| --- | --- |
| Development platform | Arduino |
| Processor | mega2560 andAVR 8bit microcontroller |
| Operating System | Ubuntu20.04 LTS |
| Programming Environment | Python3.8, Cuda11.4, Opencv4.1.1 |

### 3.1. Performance Validation under Ploughing Viewpoint

When ploughing with a tractor, the camera used for obstacle detection is often tilted, typically about 10 degrees to the left or right. This model has been improved to address the detection effect when tilted. We designed a control experiment to demonstrate that the model performs similarly well under tilted viewing angles compared to normal viewing angles. The detection results under the main camera view angle are shown in Figure 11, and it can be seen from the comparison graph that the detection performance at the tilted angle is comparable to that at the normal angle. This test result proves that even when the tractor is tilted, the model can maintain sufficient detection accuracy.

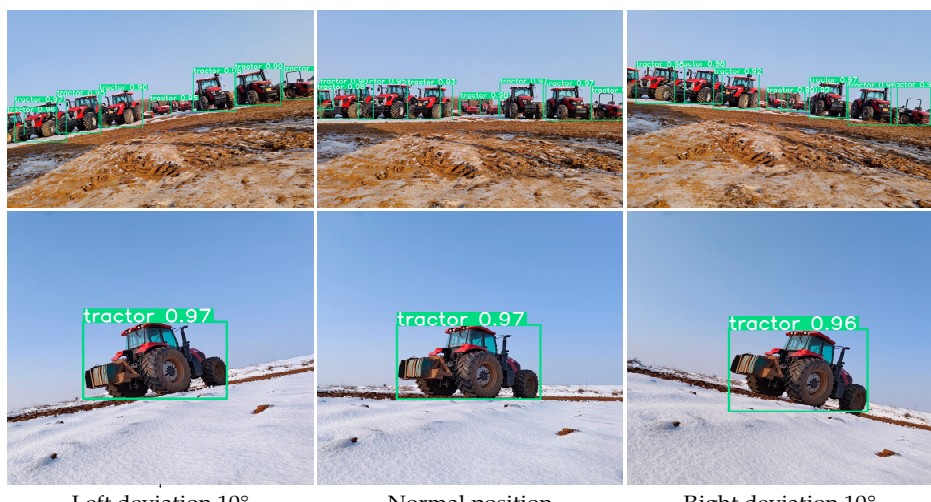

**Figure 11.** Contrast Chart of Tilted View.

*3.2. Recognition Results from Different Angles of Obstacles*

The field environment is diverse and complex, with obstacles potentially appearing at any angle within the field of view of the unmanned tractor. They may be facing away from the camera or positioned sideways. The original model's performance is unstable; when obstacles appear with their backs facing the camera's recognition range, the recognition confidence decreases significantly. To verify that our model can accurately recognize obstacles from any angle, we conducted dedicated experiments. Using the vehicle-mounted camera, we captured images of obstacles from four different angles: frontal, frontal side, lateral, and rear. These captured image samples were used as the test dataset for comparative analysis. We then employed both the original YOLOv8 model and our improved model to identify the obstacles in these images, corresponding to experiments (a) and (b) respectively, as shown in Figure 12.

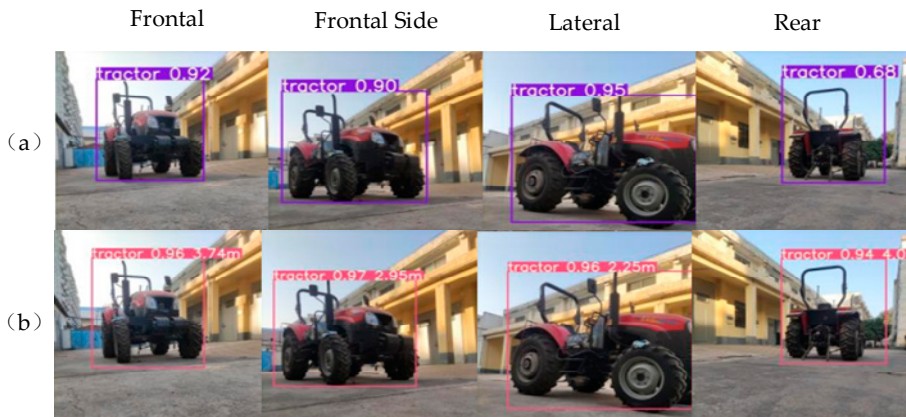

**Figure 12.** (**a**) YOLOv8 Model Recognition Performance; (**b**) Recognition Performance of Our Model.

From experiment (a), it is evident that the original YOLOv8 model achieves high recognition accuracy of over 90% when identifying obstacles from the frontal, frontal side, and lateral angles. However, when identifying obstacles from the rear angle, the recognition accuracy significantly drops to just 60%. Conversely, in experiment (b), our model demonstrates excellent recognition performance from all angles, maintaining detection accuracy of over 90% for each angle. This indicates that the improved model exhibits more stable performance and better meets the requirements of the complex field environment.

*3.3. Dynamic Verification*

While the tractor is in motion, obstacles also move within the camera's field of view. Moving obstacles can decrease detection accuracy. To verify whether recognition accuracy is significantly affected during movement, we conducted dynamic validation experiments. In the Ubuntu system, the camera of the unmanned vehicle is utilized to initiate obstacle detection and ranging through commands. Images of the vehicle were captured while the unmanned cart was controlled using a remote control handle to simulate the maximum ploughing speed of the tractor. The main camera view is shown in Figure 13.

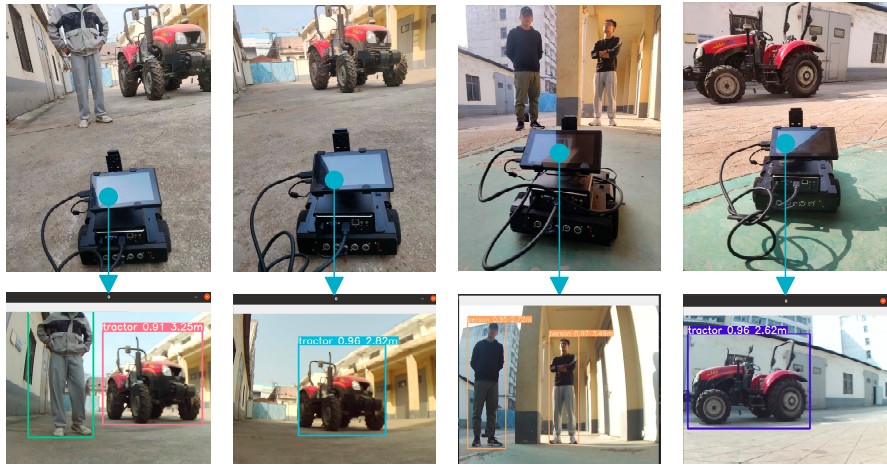

**Figure 13.** Real-time monitoring.

The test results were that the model detection confidence remained above 90% even when the robotic vehicle was in motion. This indicates that when the model is mounted on an unmanned tractor in the future, it can maintain high detection accuracy even when the tractor is ploughing forwards, and the model meets the requirement of real-time monitoring of unmanned tractor operation under normal conditions.

**4. Results**

The enhancements proposed in this paper aimed to address critical challenges in obstacle recognition for agricultural vehicles, particularly tractors operating under ploughing conditions. The improved YOLOv8 model, augmented with novel techniques, demonstrated significant advancements in accuracy, robustness, and computational efficiency.

1    Model Enhancements:

Tilted View Training Set: Introducing a specialized dataset for tilted view training enabled the model to better detect obstacles even under challenging conditions, such as when one side of the tractor's wheels is stuck in soft soil after ploughing.

FunnelReLU Activation Function: Replacing the traditional SiLU activation function with FunnelReLU significantly improved the model's robustness, particularly in scenarios with violent tractor shaking, resulting in a remarkable reduction in recognition errors for tilted obstacles.

Attention Mechanism: Embedding an SE attention mechanism enhanced the utilization of different channels in the model, further improving obstacle detection accuracy.

2    Lightweight Architecture:

MobileNetV2 and BiFPN Integration: By integrating MobileNetV2 into the backbone network and BiFPN into the neck network, the model achieved substantial reductions in computational complexity without compromising performance. This led to a 45.53% reduction in model computation, consequently saving computational resources and hardware costs.

3    Real-Vehicle Validation:

High Detection Accuracy: Real-vehicle validation experiments demonstrated the model's ability to maintain high detection accuracy even when the tractor operated with a tilted viewpoint.

Real-Time Performance: The model exhibited impressive real-time performance, achieving a detection frame rate of 67FPS during simulations of maximum ploughing speed. The average recognition accuracy surpassed 97%, with negligible error compared to simulation results.

## 5. Discussion

The findings of this study underscore the transformative impact of the proposed enhancements on agricultural vehicle assisted driving systems. Through a synergistic integration of cutting-edge technologies, including deep learning, attention mechanisms, and lightweight architectures, substantial progress has been made in bolstering the accuracy and efficiency of obstacle-recognition algorithms.

1    Advancements in Challenging Environments:

The tailored training approach for tilted perspectives, coupled with the robust FunnelReLU activation function, effectively mitigated recognition errors arising from adverse conditions such as tilting and vehicle vibrations. Furthermore, the incorporation of attention mechanisms facilitated nuanced feature extraction, thereby enhancing detection accuracy across diverse environmental contexts.

2    Optimized Computational Efficiency:

The strategic adoption of MobileNetV2 and BiFPN modules not only alleviated computational burdens but also optimized hardware resource utilization, rendering the model more adaptable to resource-constrained deployment scenarios. The transition from LIDAR to binocular ranging not only reduced computational complexity but also minimized hardware dependencies, offering a cost-effective solution for obstacle detection in agricultural settings.

3    Real-World Deployment and Future Prospects:

During tractor farming operations, which typically take place on clear days, the performance of cameras can be fully utilized, thus saving costs by relying solely on cameras for ranging. However, in practical scenarios, we still encounter issues such as camera view obstruction due to field dust and insufficient ranging precision. These situations require the integration of additional sensors such as LiDAR for multisensor fusion. Future research efforts will further refine the visual model and integrate other sensors to develop a more comprehensive obstacle detection system.

In summary, the proposed enhancements represent significant strides in advancing the effectiveness and versatility of agricultural vehicle assisted driving systems. By addressing critical challenges and harnessing state-of-the-art technologies, the model holds immense promise for enhancing operational efficiency, safety, and productivity in agricultural practices.

**Author Contributions:** Conceptualization, L.X.; methodology, X.Y.; software, B.C.; validation, Y.Z., B.C. and X.Y.; formal analysis, X.Y.; resources, M.L.; data curation, B.C.; writing—original draft preparation, B.C.; writing—review and editing, X.Y. and Y.Z.; supervision, L.X.; project administration, X.Y.; funding acquisition, M.L. All authors have read and agreed to the published version of the manuscript.

**Funding:** This research was funded by the following grants: Henan Provincial Key R&D Special Project (Grant No. 231111112600), 14th Five-Year National Key R&D Programme (Grant No. 2022YFD2001201B), Henan Provincial Universities Scientific and Technological Innovation Team Supporting Scheme Project (Grant No. 24IRTSTHN029), and Open Subject of the National Key Laboratory

of Intelligent Agricultural Power Equipment (Grant No. SKLIAPE2023006). The article processing charge was supported by the Henan Provincial Key R&D Special Project (Grant No. 231111112600).

**Data Availability Statement:** The data presented in this study are available on request from the corresponding author.

**Conflicts of Interest:** The authors declare no conflict of interest.

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
