# Peer review of "Inclined Obstacle Recognition and Ranging Method in Farmland Based on Improved YOLOv8"

_wevj, doi:10.3390/wevj15030104_

Round 1
Reviewer 1 Report
Comments and Suggestions for Authors
Dear Authors,
The article presents a model to complement real-time monitoring of obstacles in the area of unmanned tractors using a camera based on the YOLOv8 network model. Various aspects of this model, experimental tests and verification of the system's operation are presented.
The manuscript requires some improvements:
1. There are many editing and linguistic errors, unfinished sentences and too long complex sentences. In addition, the formatting of the work should be improved.
2. In line 28 you can supplement the references with additional items [1; ...]:
https://doi.org/10.26552/com.C.2016.2.30-33
https://doi.org/10.1051/matecconf/201824403005
3. Lines 81-91, this fragment should be reworded, the sentences are too complex.
4. Similarly in subsection 1.1.
5. Add a caption in Table 1.
6. Line 281 where is table 4?
7. The manuscript lacks a comparison (discussion) of the results obtained by the authors with other works in this field. This can be compared to differences from tests in motor vehicles.
In general, the manuscript requires improvement in the organization of the work, as well as the rewording of too complex sentences, the ordering of the results and discussion with other works in this area.
Thank you!
Comments on the Quality of English LanguageThere are minor editing and stylistic errors that need to be corrected, e.g.: Line 49; 53-54 etc. The sentences are too long and complex, e.g. line 81-91.
Author Response
Thank you very much for your review comments, which are very important to us. We have made changes and replied to each of your review comments, please guide us, thank you.
Response to Question 1: Thank you for your comments, I have made changes to some statements in the article.
Response to Question 2: Thank you, the articles you listed are very useful for the enhancement of this article and I have added them to my references.
Response to Question 3: The paragraph has been revised to make it more concise.
Response to Question 4: Changes have also been made to some of the difficult statements in Section 1.1.
Response to Question 5: I'm sorry, I was really careless and have made changes.
Response to Question 6: Sorry, it should be Form 2, I was too careless.
Response to Question 7: Sorry, the comparison with motorised vehicles was not made as the model was applied to a tractor.
Response to Question 8: I have made changes to line 49, 53-54 you mentioned, including other sentences that did not make sense.
Thank you for your valuable comments!
Reviewer 2 Report
Comments and Suggestions for Authors
This paper enhances the YOLOv8 model through some key innovations: a custom tilted-view training dataset, a novel FunnelReLU activation function for heightened sensitivity to irregular obstacles, and an integrated attention mechanism.
The paper is nice and interesting; however, I have several concerns:
1. The citations should not be in superscript, but rather in the regular font.
2. In Figure 2, the author needs to specify what information goes through each arrow drawn in the figure.
3. The authors write "As shown in Fig. 3(a), the confidence level of the original model is 96% at the highest and 46% at the lowest". This is a huge difference, but no explanation for this huge difference is given. It is worth adding an explanation.
4. In equations 1,2,3, what are I and j?
5. In Figure 4 there is no explanation for the matrices to which the authors compare. What are these matrices?
6. In Table 1, more explanation is needed. E.g. What is t?
7. In Figure 5, the word "squeeze" usually focuses on reducing the physical size or volume of something. The word "compress" seems to be more suitable.
8. In Figure 9, what is the label of the y axis?
9. The equations and the results seem to be detached. Please explain how you have designed the experiments based on the theoretical background.
10. In the conclusions, the authors write "adopt binocular ranging instead of LIDAR". Why instead of LIDAR and not in addition to LIDAR? In Y. Wiseman, "Ancillary Ultrasonic Rangefinder for Autonomous Vehicles", International Journal of Security and its Applications, Vol. 12(5), pp. 49-58, 2018 available online at: https://u.cs.biu.ac.il/~wisemay/ijsia2018.pdf , and also in Yee, P. D. R., Pinrath, N., & Matsuhira, N. (2020, September). "Autonomous mobile robot navigation using 2D LiDAR and inclined laser rangefinder to avoid a lower object", In the 2020 59th Annual Conference of the Society of Instrument and Control Engineers of Japan (SICE), pp. 1404-1409, the authors suggest adding also Ultrasonic Rangefinder to the LIDAR. I would encourage the authors to cite these papers and to explain how their binocular ranging can be added to LIDAR with the aim of creating a better rangefinder at least as a future work.
11. The paper only emphasizes the advantages of the work done, but does not elaborate on the shortcomings of the work done and the follow-up prospects.
12. The format of references should be consistent.
Author Response
Thank you very much for your review comments, which are very important to us. We have made changes and replied to each of your review comments, please guide us, thank you.
Response to question 1: I am sorry, this was my mistake and I have made the changes.
Answer to Question 2: Hi, I just used arrows to indicate the information transfer route, and in the binocular ranging module, I marked the information transferred by each arrow, but in the YOLOv8 algorithm, due to the lack of space in the graph, if each arrow is added with the information transferred, then there are too many contents and no place to put them.
Response to Question 3: Hello, I would like to express that this is due to the stability of the SiLU activation function is not good enough, in the picture appears slightly tilted, the recognition effect will be unstable, in contrast, because this model uses the FunnelReLU activation function, so that the model is more resistant to interference, even in the case of the tractor in the ploughing of the camera is often jittery, the recognition effect of the obstacle will will not change greatly. This explanation was also added to the original paper.
Response to question 4: i and j indicate the coordinate position of the centre window, and this explanation has also been added to the original text.
Response to question 5: Sorry, this was a mistake on my part and the explanation of the matrix has been added.
Response to question 6: Sorry, this was my mistake, explanation has been added to the original text
The "t" stands for "expansion factor", which is used to control the multiplicity of expansion. In MobileNetV2, each residual block has a deep expansion." The "t" value indicates the number of input channels relative to the number of output channels during expansion. If the "t" value is 1, then no channel expansion is performed.
"c" indicates the number of output channels, i.e., the depth of the feature map output by each residual block.
"n" denotes the number of repetitions (repeats), i.e., the number of times a residual block of that type is reused throughout the network.
Response to Question 7: Thank you for your valuable comments, changes have been made.
Response to Question 8: The y-axis of the left graph is the confidence level and the y-axis of the right graph is the loss value, which has also been modified in the original article.
Response to Question 9: We are trying to apply the environmental sensing technology on the unmanned vehicle to the tractor by combining it with the tractor's working characteristics.
Response to Question 10: Hi reviewer, the reason why we said to replace LIDAR instead of using it in combination with LIDAR is to save cost. Considering the cost of agricultural machinery, the idea is to use a camera instead of LiDAR.
Response to Question 11: It has been added on, the outlook of the work going forward and the lack of work.
Response to Question 12: OK, thanks for the reminder.
Thank you for your valuable comments and have a nice life!
Reviewer 3 Report
Comments and Suggestions for Authors
The topic of the paper could be interesting , although the paper must be better framed in the literature state-of-art and the novelty of the paper must be highlighted compared to literature:
- In the introduction, the novelty of the paper is not mentioned; please improve this aspect;
- The research gap covered by this study was not highlighted with comparison to literature;
- In the section 2.4 "Comparison test" you mention "the performance of the improved model" , what the improved model consist in? It is not clear what is the novel aspect;
- Until section 2.4, it seems you only used an already existing software tool, what have you done more and have you added to it?
- In the conlusion , you say that the acuracy of the model is improved by 2.34% and the recall rate is improved by 2.95% , how did you get this performance? Where and how did you compare your method with exisitng literature?
Comments on the Quality of English Language
The english of the paper is good.
Author Response
Thank you very much for your review comments, which are very important to us. We have made changes and replied to each of your review comments, please guide us, thank you.
Response to Question 1: Thank you for your comments, I have made changes in the article.
Reply to Question 2: The research of this paper mainly focuses on combining the environmental sensing technology with the working characteristics of tractor ploughing, which makes the obstacle recognition technology more suitable to be applied to the tractor.
Answer to Question 3: Hello, the improved model refers to the model in this paper, which has been changed to this model to avoid misunderstanding.
Response to Question 4: Hi, in addition to the algorithm improvement, I have added the binocular ranging mechanism.
Response to Question 5: There is a table file containing the training report generated after the model training is completed. After training this model, the table file shows that the mAP value of this model is 98.84% and the recall is 95.81%. While training the original YOLOv8 model the mAP value is 96.50% and the recall is 92.86%.
Hello, I have compared with the existing literature in the introduction, and nowadays there are very few environmental sensing techniques combined with tractor working conditions for field environment obstacle recognition.
Thank you for your valuable comments and have a nice life!
Reviewer 4 Report
Comments and Suggestions for Authors
This work proposes a new method for recognizing inclined obstacles based on an improved YOLOv8 model, for farmland activities. There are several issues:
1. Section 1.2. It is not clear how the latitude determines the computational resources.
2. Fig 1. What mean t, c, n?
3. Section 1.3. Eqs must start from 4, not from 7. What mean w, g, uc? Please give an example of a combination operation z.
4. Section 1.5. Fig 7 is included in Fig 2. In the text, the parallax is determined by Xr and Xt. In Fig 7, there are XR and XT.
5. Section 2.3. What mean ri, ri+1, Pinter? FP is wrongly defined: it is the number of negative categories misjudged. The Recall is wrongly defined, too.
6. Section 2.4. What is the leakage rate?
7. Conclusion. Please specify that the accuracy is improved for the used dataset, not in general.
Comments on the Quality of English LanguageThere are several mistakes. For example:
- debasis Kumar
- YOLO algorithm's vehicle detection accuracy;
- Fig 2 comes after an unfinished sentence. Also, the caption of this figure misses some capital letters.
- Frsmes persecond.
- and others.
Author Response
Thank you very much for your review comments, which are very important to us. We have made changes and responses to each of your review comments, please guide us, thank you.
Response to Question 1: Sorry, I asked a professional and the result is So expanding the number of channels does not add a lot of extra computation, expanding the dimension is to increase the expressive power of the network. By expanding the number of channels through a 1x1 convolutional layer, more feature combinations and interaction information can be introduced, thus increasing the expressive power of the network. This allows the network to better extract and capture features in the image during the depth-separable convolution stage. Finally by shrinking the number of channels back by another 1x1 convolutional layer, a smaller number of parameters and computation can be maintained while preserving the information from the original input. So expanding the number of channels doesn't add a lot of extra computation, but can help the network learn and represent image features better.
I have also made changes in the original article.
Response to Question 2: Sorry, it was an oversight on my part, and I have now added the explanation in the original article.
"t" stands for "expansion factor", which is the expansion factor, used to control the multiplicity of expansion (expansion). In MobileNetV2, each residual block has a deep expansion component." The "t" value indicates the number of input channels relative to the number of output channels during expansion. If the "t" value is 1, then no channel expansion is performed.
"c" indicates the number of output channels, i.e., the depth of the feature map output from each residual block.
"n" denotes the number of repetitions (repeats), i.e., the number of times a residual block of that type is reused throughout the network.
Response to Question 3: Sorry, the equation numbering has been changed, W denotes the weights, and "U" in Uc is a vector representing the low-dimensional embedding of the feature map obtained through the global average pooling operation. The length of this vector is CC, the number of channels in the input feature map. g is a hyperparameter that regulates the dimensionality change between fully connected layers.
Response to Question 4: Sorry, this was my mistake and the image has been corrected.
Response to Question 5: ri and ri+1ri+1 are the recall values corresponding to the first interpolated value in the precision interpolation stage. These values are in ascending order and are used to compute the interpolation between precision and recall. pinterPinter is the interpolated value in the precision interpolation stage and is used to interpolate between different recall values. The definitions of FP and FN are wrong. fp denotes the number of negative categories that were misclassified and fn denotes the number of positive categories that were misclassified. Recall is the proportion of the number of samples correctly identified as positive to the number of samples in all positive categories, i.e., TPTP+FNTP+FNTP. precision is the proportion of the number of samples correctly identified as positive to the number of samples in all positive categories, i.e., TPTP+FPTP+FPTP. mAP is the mean precision value for all categories. categories' average precision values.
Response to Question 6: It is the probability that a sample is not detected. It has been replaced with false negative rate in the original text.
Question 7, thanks for your suggestion, I have corrected these grammatical errors.

Round 2
Reviewer 1 Report
Comments and Suggestions for Authors
Dear Authors,
thank you for the changes made and satisfactory responses to the comments from the review. The manuscript is much better than the original version, so I recommend it for printing in its current form.
Regards
Author Response
thank you.
Reviewer 2 Report
Comments and Suggestions for Authors
First of all, when submitting an article with corrections, the changes must be marked because otherwise it is difficult to see what was done and where. It also appears in the instructions that MDPI sends out when a revision is needed.
Remark 9, Can you please add this explanation to the text?
Remark 10, I can understand that you want to save cost, but it is worth mentioning that in the future the financial consideration might be different and then it will be worth considering adding the Lidar as an add-on equipment.
I did not find some of the changes and additions that the authors claim they added or changed, but maybe that is because they do not indicate where the changes are and it is hard to go through the entire text, especially since they do not mark the changes and additions.
Author Response
Dear Reviewer:
Hello! We deeply apologize for not annotating the changes made to the original manuscript. We have now highlighted all the modifications in yellow in the text according to your feedback from the first and second rounds of review. Additionally, we have added annotations for each change. We hope you will review them. Thank you.
(1)In response to your Comment Nine: It appears there might be a disconnect between the equations and the results. We deeply apologize for this oversight. Upon reflection, we recognize that there may be insufficient coherence between the experimental design and the results in the article, which requires further refinement and integration. To address this issue, we have made the following improvements. We have added language in Sections 3.1, 3.2, and 3.3 explaining the experimental purposes, and we have rewritten Section 4 to make it easier for readers to understand the purpose, methods, and results of the study. These improvements are annotated in the text and highlighted in yellow for your reference. Thank you.
(2)In response to your Comment 10 regarding the issue of cameras not replacing LiDAR, we offer the following reply. Initially, our intention was to capitalize on the performance of cameras during tractor farming operations, which typically occur in clear daylight, to save costs by solely relying on cameras for ranging. However, in practical scenarios, we still encounter challenges such as camera view obstruction due to field dust and inadequate ranging precision. These situations necessitate the integration of additional sensors like LiDAR for multisensor fusion. Future research endeavors will focus on further refining the visual model and integrating other sensors to develop a more comprehensive obstacle detection system.We have made changes and annotations in the third paragraph of Section 5 in the manuscript to address this issue. We hope you will review it. Thank you.
In response to the feedback from the first round of review, we have made annotations and highlighted changes in yellow throughout the manuscript for your reference. Thank you.
Thank you once again for taking the time to review our manuscript and provide valuable feedback. We greatly appreciate your insights and have carefully considered and implemented the necessary revisions accordingly. Should you have any further questions or suggestions, please do not hesitate to contact us. We look forward to potential future collaborations. Wishing you all the best, thank you!
Best regards.

Reviewer 3 Report
Comments and Suggestions for Authors
The authors improved the paper according to the reviewer's comments.
Comments on the Quality of English Language
The english is ok.
Author Response
thank you for your reviewing.
Round 3
Reviewer 2 Report
Comments and Suggestions for Authors
The authors made a decent effort and the paper is certainly publishable so I would recommend accepting the paper.